# Quercetin Reprograms Immunometabolism of Macrophages via the SIRT1/PGC-1α Signaling Pathway to Ameliorate Lipopolysaccharide-Induced Oxidative Damage

**DOI:** 10.3390/ijms24065542

**Published:** 2023-03-14

**Authors:** Jing Peng, Zhen Yang, Hao Li, Baocheng Hao, Dongan Cui, Ruofeng Shang, Yanan Lv, Yu Liu, Wanxia Pu, Hongjuan Zhang, Jiongjie He, Xuehong Wang, Shengyi Wang

**Affiliations:** Key Laboratory of New Animal Drug Project, Gansu Province, Key Laboratory of Veterinary Pharmaceutical Development, Ministry of Agriculture and Rural Affairs, Lanzhou Institute of Husbandry and Pharmaceutical Sciences of Chinese Academy of Agriculture Sciences, Lanzhou 730050, China

**Keywords:** quercetin, inflammation, oxidative stress, macrophage, redox, immunometabolism, SIRT1/PGC-1α signaling pathway

## Abstract

The redox system is closely related to changes in cellular metabolism. Regulating immune cell metabolism and preventing abnormal activation by adding antioxidants may become an effective treatment for oxidative stress and inflammation-related diseases. Quercetin is a naturally sourced flavonoid with anti-inflammatory and antioxidant activities. However, whether quercetin can inhibit LPS-induced oxidative stress in inflammatory macrophages by affecting immunometabolism has been rarely reported. Therefore, the present study combined cell biology and molecular biology methods to investigate the antioxidant effect and mechanism of quercetin in LPS-induced inflammatory macrophages at the RNA and protein levels. Firstly, quercetin was found to attenuate the effect of LPS on macrophage proliferation and reduce LPS-induced cell proliferation and pseudopodia formation by inhibiting cell differentiation, as measured by cell activity and proliferation. Subsequently, through the detection of intracellular reactive oxygen species (ROS) levels, mRNA expression of pro-inflammatory factors and antioxidant enzyme activity, it was found that quercetin can improve the antioxidant enzyme activity of inflammatory macrophages and inhibit their ROS production and overexpression of inflammatory factors. In addition, the results of mitochondrial morphology and mitochondrial function assays showed that quercetin could upregulate the mitochondrial membrane potential, ATP production and ATP synthase content decrease induced by LPS, and reverse the mitochondrial morphology damage to a certain extent. Finally, Western blotting analysis demonstrated that quercetin significantly upregulated the protein expressions of SIRT1 and PGC-1α, that were inhibited by LPS. And the inhibitory effects of quercetin on LPS-induced ROS production in macrophages and the protective effects on mitochondrial morphology and membrane potential were significantly decreased by the addition of SIRT1 inhibitors. These results suggested that quercetin reprograms the mitochondria metabolism of macrophages through the SIRT1/PGC-1α signaling pathway, thereby exerting its effect of alleviating LPS-induced oxidative stress damage.

## 1. Introduction

The oxidation–reduction (redox) system plays an important role in maintaining cellular redox homeostasis and preventing oxidative damage to DNA, proteins and lipids caused by free radicals. The latest research suggest that the redox system is closely intertwined and interdependent with changes in cellular metabolism. Activated immune cells undergo specific metabolic reprogramming and participate in the regulation of the redox system, which ultimately determines the function of immune cells [1]. The level of cellular oxidative stress can be effectively reduced by modulating the immune metabolism of cells [1,2,3]. Therefore, reducing the hyperresponsiveness of the innate immune system, by regulating immune metabolism with the addition of antioxidants, can reduce the damage to the organism caused by oxidative stress due to overactivation of the immune system.

Inflammation is the host’s natural defense response to pathogens or tissue damages and is important for elimination of harmful stimuli and initiation of the healing process [4]. Lipopolysaccharide is one of the major inflammation-causing substances. It stimulates the production of reactive oxygen species (ROS) by macrophages and infiltrating neutrophils through a variety of mechanisms, including activation of NADPH oxidase and inhibition of antioxidant enzymes involved in ROS scavenging [5,6]. ROS act as intracellular messengers that activate redox-sensitive transcription factors, such as nuclear factor κB, thereby stimulating the production of pro-inflammatory cytokines [7,8]. The release of these cytokines may trigger the production of ROS by non-phagocytes [9]. Excessive production of ROS may lead to cellular and tissue damage and trigger many metabolic diseases associated with chronic inflammation and oxidative stress, such as neurodegenerative diseases, cardiovascular diseases, etc. [10,11].

Macrophages are heterogeneous cells that play a key role in inflammatory and tissue repair responses. Changes in cellular metabolism are important determinants of macrophage function and phenotype [12]. It has been reported that cellular requirements in specific microenvironments (e.g., inflamed tissues, tumors, etc.), such as survival, growth and proliferation, or specific effector functions, such as phagocytosis execution and cytokine production, would be met by reprogramming the metabolic phenotype. The function of immune cells, especially macrophages, has the potential to assist in the treatment of diseases with a high macrophage load by regulating metabolism. Therefore, it is a promising approach for the treatment of oxidative stress and inflammation-related diseases by metabolic reprogramming of macrophages, such as cancer, atherosclerosis, etc. [13,14]. Macrophages play the critical and plastic role in host defense, immune regulation and wound healing by adapting to the local environment and adopting different phenotypes [15]. Generally, macrophages are divided into two main phenotypes based on different activation pathways that are the “classically activated”, pro-inflammatory M1 type, and the “alternatively activated”, anti-inflammatory M2 type. Macrophages in different activation states have different metabolic characteristics and are involved in a variety of pathophysiological processes [16]. Macrophages can cause a variety of diseases when the complex balance of macrophage activity is disrupted [15]. For example, the important relationship between specific steps in the development of colon cancer and obesity-induced inflammation has been pointed out to be mediated by inflammatory cytokines secreted by macrophages, such as interleukin 6 (IL-6) and tumor necrosis factor-alpha (TNF-α) [17,18]. In addition, static macrophages undergo pro-inflammatory differentiation in response to bacteria and LPS characterized by the release of large amounts of cytokines, such as TNF-α, interleukin 1β (IL-1β), IL-6, as well as ROS [19]. These pro-inflammatory mediators are essential for the immune defense of organisms and the killing of microorganisms. However, sustained inflammatory activation of macrophages may lead to collateral tissue damage and chronic inflammation. Therefore, preventing abnormal activation of macrophages and keeping them in the appropriate range of activation is essential to avoid diseases related to inflammatory activation caused by macrophages.

Flavonoids have been reported to possess antioxidant, anti-inflammatory, anti-allergy, anti-mutation and anti-cancer effects [20,21,22,23,24]. Studies have shown that flavonoids exert antioxidant effects mainly through scavenging ROS, chelating metal ions, regulating GSH levels, increasing the expression of endogenous antioxidant enzymes, improving the antioxidant properties of low-molecular antioxidants, and regulating various signal transduction pathways, such as the nuclear factor erythroid 2-related factor 2 (NRF2) pathway [21,25,26]. This redox-sensitive transcription factor promotes the induction of phase II detoxification enzymes and other cytoprotective proteins by binding to antioxidant response elements (ARES). NRF2 has been demonstrated to be an important regulatory factor of the inflammatory response. Peritoneal macrophages of NRF2-deficient mice exhibited the increased expression of binding activity genes of proinflammatory cytokines and NF-κB after administration of bacterial endotoxin. Numbers of natural flavonoids, including quercetin, kaempferol, paclitaxel and proanthocyanidins, have been identified as NRF2 activators and therefore show potential for the treatment of certain diseases, such as inflammation, cancer and cardiovascular disease [27].

Quercetin is a naturally occurring flavonoid that is widely found in a variety of fruits and vegetables, including blueberries, broccoli, onions, green tea, etc. It has been illustrated to be one of the most active scavengers of ROS and RNS in vitro and in vivo [21,28,29]. The antioxidant activity of quercetin has been reported to not only prevent oxidative stress, but also reduce inflammation by blocking the ROS-inflammatory cycle [30]. Furthermore, quercetin can alleviate acute lung injury by reducing the levels of oxidative stress markers and increasing the activity of antioxidant enzymes [31]. In the LPS-induced rat model, the levels of superoxide dismutase and catalase increased, and the levels of malondialdehyde decreased after treatment with quercetin, indicating that quercetin enhanced the antioxidant defense system of the rat model [32]. Researchers have conducted studies on the alleviating effect of quercetin on diseases characterized by altered immune response. For example, GSH has been indicated to be involved in the antioxidant process of quercetin’s inhibition of endotoxin-induced oxidant production by human aortic endothelial cells (HAEC) [33]. Activation of bone-marrow-derived dendritic cells and production of cytokines could be reduced by quercetin [34]. The renal inflammation and functional impairment caused by cisplatin could be significantly reduced by quercetin [35]. Alongside that, due to the natural compound properties of quercetin, it is reasonable to predict that it can exert its pharmacological effects with minimal side effects. This was also confirmed by the results of the study on the gene protective and cytoprotective effects of quercetin [36]. Therefore, quercetin is a promising drug for clinical application. However, few studies have reported whether quercetin can inhibit oxidative stress in LPS-induced inflammation by influencing the immune metabolism.

Herein, the purpose of this study was to determine the antioxidant effect and mechanism of quercetin in LPS-induced macrophage inflammation. Firstly, the effects of quercetin on the viability and proliferation of LPS-induced RAW264.7 cells were evaluated. Then, the relationship between quercetin and ROS production, mRNA expression of pro-inflammatory cytokines, antioxidant enzyme activity and mitochondrial function in inflammatory macrophages was investigated. Further, the effect of quercetin on the protein expression levels of SIRT1 and PGC-1α in each treatment group was examined by Western blotting. In addition, the SIRT1 inhibitor EX527 was used to explore whether the protective effect of quercetin was related to SIRT1. The results suggested that quercetin may exert its protective effect against oxidative stress in LPS-induced inflammatory macrophages through the SIRT1/PGC-1α signaling pathway.

## 2. Results

### 2.1. Quercetin Increases the Viability and Decreases the Differentiation of LPS-Induced RAW264.7 Cells

To assess the role of quercetin in LPS-induced macrophages, we first evaluated the effect of quercetin on RAW264.7 cell viability. Cells were treated with 0 to 160 µM of quercetin for 24 h. Cell viability was measured using an MTT assay, which is commonly used for assessing the viability of cells and measuring the cytotoxicity of compounds. As shown in Figure 1B, the cell viability was significantly decreased compared to the blank control group when the concentration of quercetin reached 20 µM (*p* < 0.01). This indicated that quercetin was not toxic to cells in the concentration range of 2.5 to 10 µM, while it showed cytotoxicity at 20 µM. Therefore, 10 µM was considered as the maximum non-cytotoxic dose of quercetin and was selected for the subsequent experiments. In addition, 1 µg/mL of LPS was used to induce a cellular oxidative stress model, based on the literature reported [37].

Interestingly, although many types of adherent cells are known to detach from the surface during cell death, macrophages appear to remain strongly attached to the plate even after substantial morphological changes and loss of viable function. As shown in Figure 1C, LPS-treated RAW264.7 cells exhibited significant morphological changes compared to the blank control group, most of which were irregular, and these changes were ameliorated by quercetin. Our results suggested that quercetin can reduce LPS-induced cell spreading and pseudopodia formation by inhibiting cell differentiation. Since the cell morphological changes were obtained by LASEZ microscope image software-assisted imaging, we sought to provide clearer evidence to support these findings.

An EdU assay was used to determine the effect of quercetin in the proliferation of LPS-induced RAW264.7 cells. As illustrated in Figure 1D,E, the number of EdU-positive cells was significantly reduced by 93% in LPS group compared to the blank control group (*p* < 0.01), while the number of EdU-positive cells in the quercetin-treated group was 5.13 times higher than that in LPS group (Figure 1D,E, *p* < 0.05), suggesting that quercetin attenuated the effect of LPS on macrophage proliferation. The above data support the protective effect of quercetin against LPS-induced macrophage damage, cell morphological changes and cell proliferation reduction.

### 2.2. Quercetin Inhibits ROS Production and Pro-Inflammatory Cytokine Expression, and Improves the Antioxidant Capacity of LPS-Induced RAW264.7 Cells

ROS is the key executor in cellular and tissue damage caused by oxidative stress, and is closely associated with diseases [38]. The fluorescent probe (DCFH-DA) was employed to determine the effect of quercetin on ROS production. Compared to the blank control group, the fluorescence intensity of DCF (the oxidation product of DCFH-DA) in the LPS group was significantly increased by 1.08 times (*p* < 0.01). In contrast, the fluorescence intensity of DCF in quercetin-treated group was significantly reduced by 0.25 times compared to the LPS group (*p* < 0.01, Figure 2A,B). These results suggested that quercetin has an inhibitory effect on LPS-induced ROS production in macrophages.

For further understanding of the antioxidant capacity of quercetin, the effects of quercetin on the enzyme activity in LPS-induced RAW264.7 cells were evaluated. As presented in Figure 2C,D, LPS significantly reduced the level of GSH by 37% compared to the blank control group (*p* < 0.05), while the quercetin-treated group showed a significant 0.82-fold increase in the level of GSH compared to the LPS group (*p* < 0.01). The opposite results were observed at the MDA level, that was, LPS significantly increased the level of MDA by 41% compared to the blank control group (*p* < 0.01), while the quercetin-treated group showed a significant decrease of 32% in MDA level (*p* < 0.01). The results indicated that quercetin could exert its antioxidant effect by increasing intracellular GSH levels and decreasing MDA levels.

To verify the role of quercetin in the pro-inflammatory cytokine expression, the mRNA levels of TNF-α, IL-6, IL-1β and NF-κB in LPS-induced RAW264.7 were determined by quantitative real-time PCR. As demonstrated in Figure 2E–H, the mRNA expression of TNF-α, IL-6, IL-1β and NF-κB showed similar trends. That was, LPS significantly increased their mRNA expression compared to the blank control group, with a fold increase of 17.21, 57,143, 2222.76 and 0.87, respectively (*p* < 0.01). On the other hand, the high expression of these pro-inflammatory cytokines was significantly decreased by 31%, 56%, 35% and 41%, respectively, after treatment with quercetin (*p* < 0.01). Results indicated that quercetin can reduce the high expression of inflammatory cytokines caused by LPS, and thus inhibit the inflammatory response.

### 2.3. Quercetin Protects Mitochondrial Function and Prevents LPS-Induced Mitochondrial Morphological Damage

Mitochondrial membrane potential is one of the indicators of mitochondrial function. The dysfunction of mitochondrial morphological potential in LPS-induced RAW264.7 cells was detected by JC-1 staining in this study. As shown in Figure 3A, cells in the blank control group and quercetin group exhibited strong red fluorescence under inverted fluorescence microscope, while the intensity of red fluorescence of cells in the LPS group was significantly decreased. Notably, the red fluorescence intensity of the quercetin-treated group was significantly enhanced compared to that of the LPS group. The results of JC-1 ratio analysis (Figure 3B) showed that the mitochondrial membrane potential in the LPS group was significantly reduced by 48% compared to the blank control group (*p* < 0.01), while that of the quercetin-treated group was significantly increased by 49% compared to the LPS group (*p* < 0.01). The experimental results demonstrated that quercetin inhibits the LPS-induced decrease in mitochondrial membrane potential, thus reducing mitochondrial damage and protecting mitochondrial function.

Further, the mitochondrial ultrastructure in RAW264.7 cells was observed by transmission electron microscopy, to visualize the effect of quercetin on LPS-induced changes in cellular mitochondrial morphology. As displayed in Figure 3C, mitochondrial morphology of the blank control group and quercetin group was normal, with round or oval shape, continuous outer membrane, tightly arranged inner cristae and uniform matrix density. However, the mitochondria of the LPS-treated cells were obviously swollen, the matrix was thin and contained a small amount of flocculent material, the matrix particles disappeared, cristae lysed and fractured, and a large number of vacuoles appeared. LPS-treated cells exhibited obvious damaged mitochondria relative to the blank control. Meanwhile, the electron microscopic images of mitochondrial morphology in the quercetin-treated group showed that some of the mitochondria in this group of cells had normal morphology, and some of the mitochondria were mildly swollen, and the intercristae space was enlarged. According to the degree of mitochondrial morphological alterations in each group, it could be deduced that quercetin protects against mitochondrial damage caused by LPS to some extent.

Mitochondrial DNA (mtDNA) damage, as well as the decline in mitochondrial RNA (mtRNA) transcription, protein synthesis, and mitochondrial function, can reflect mitochondrial damage. The possible reasons that mtDNA is more susceptible to oxidative stress damage than nuclear DNA are that it is close to the respiratory chain of the inner mitochondrial membrane, lacks protective histone-like proteins and has less repair activity for damage [39]. Mitochondrial dysfunction could be assessed by the quantification of mtDNA copy number [40]. In this study, the mtDNA copy number was expressed as the ratio of 18S RNA to mtDNA. As presented in Figure 3D, the mtDNA copy number in the quercetin group was significantly increased by 33% compared to the blank control group (*p* < 0.01), while the decrease of mtDNA copy number in the LPS group was not significant (*p* > 0.05). Notably, the mtDNA copy number of the quercetin-treated group was not significantly different from that of the LPS group, suggesting that the protective effect of quercetin against cell damage caused by LPS was not significantly related to mitochondrial dysfunction.

To further investigate the effect of quercetin on mitochondrial function, ATP content, which depends on the cellular mitochondrial activity, was measured. As illustrated in Figure 3E, treatment with quercetin alone significantly increased the ATP content by 13% (*p* < 0.01), and stimulation with LPS alone significantly decreased the ATP content by 17% (*p* < 0.01) compared to the blank control group. Alongside that, the ATP content of the quercetin-treated group was significantly increased by 15% compared to that of the LPS group (*p* < 0.01). Thus, it can be concluded that quercetin contributes to the increase the intracellular ATP production. Meanwhile, the content of ATP synthase, which is an important enzyme involved in mitochondrial oxidative phosphorylation [41], was measured to monitor the mitochondrial function in LPS-induced RAW264.7 cells. As shown in Figure 3F, the content of ATP synthase in the LPS group was significantly decreased by 25% compared to the blank control group (*p* < 0.01), whereas in the quercetin-treated group, it was significantly increased by 22% compared to the LPS group (*p* < 0.05). This is consistent with the experimental results of ATP content measurement described above. These results indicated that quercetin attenuates the adverse effects of LPS on ATP production and ATP synthase, thereby protecting mitochondrial function.

### 2.4. Quercetin Suppresses LPS-Induced ROS Production and Mitochondrial Damage in RAW264.7 Cells via the SIRT1/PGC-1a Signaling Pathway

The SIRT1/PGC-1a signaling pathway is closely related to the regulation of cellular oxidative stress and inflammation [42,43]. To investigate the mechanism by which quercetin exerts its antioxidant effects in inflammatory macrophages, the expression of SIRT1 and PGC-1α proteins was measured. Western blotting analysis showed that SIRT1 and PGC-1α protein levels in RAW264.7 cells stimulated with LPS were significantly downregulated by 25% and 31%, respectively, compared to the blank control group (*p* < 0.05, Figure 4A–C). Simultaneously, compared to stimulation with LPS alone, the protein expression of SIRT1 and PGC-1α was significantly elevated by 18% and 51%, respectively, when quercetin was added for co-treatment after LPS stimulation (*p* < 0.05, Figure 4A–C), indicating that the protection of quercetin against LPS-induced oxidative damage in macrophages may be regulated through the SIRT1/PGC-1α signaling pathway.

To further assess the role of SIRT1 in the signaling mechanism of quercetin, EX527 was administered to inhibit SIRT1. As demonstrated in Figure 4D, the fluorescence intensity of DCF was obviously enhanced in the EX527 group compared to the quercetin-treated group. Consistent with this, the intracellular ROS content of the EX527 group was 23% higher than that of the quercetin-treated group (*p* < 0.05, Figure 4E). Therefore, the inhibition of SIRT1 attenuated the inhibitory effect of quercetin on LPS-induced ROS production in macrophages. The determination of mitochondrial membrane potential showed that, compared to the quercetin-treated group, the red fluorescence intensity of the EX527 group was obviously diminished (Figure 4F) and the JC-1 ratio (*p* < 0.05, Figure 4G) was significantly decreased by 48%, indicating that inhibition of SIRT1 reversed the protective effects of quercetin on mitochondrial membrane potential. In addition, transmission electron micrographs (Figure 4H) of the mitochondrial ultrastructure showed that the mitochondria in the EX527 group were significantly swollen, with a large number of vacuoles and severe morphological damage compared to the quercetin-treated group. Thus, it can be seen that the inhibition of SIRT1 reversed the protection of quercetin on the mitochondrial morphology of inflammatory macrophages.

## 3. Discussion

Inflammation is a host defense mechanism of the organism against harmful stimuli from pathogens and is characterized by excessive production of ROS by activated immune cells (macrophages, plasma cells and lymphocytes, etc.) [44]. Adducts produced by the reaction of ROS and lipids, proteins and DNA lead to oxidative stress and induces the release of cytokines, growth factors and chemokines that stimulate pathways leading to amplified inflammation [44,45]. The cycle of inflammation can be perpetuated by ROS production and oxidative stress, which leads to a chronic state that drives a variety of inflammatory pathologies, ultimately leading to cellular damage and death [46,47]. Given the inseparable and closely linked relationship between inflammation and oxidative stress, modulating the redox balance of inflammatory immune cells and therapeutically inhibiting the inflammation–oxidative stress cycle may become an effective approach for the treatment of inflammation-related diseases. Here, we examined the effects of quercetin on the cell morphology and cell proliferation of LPS-induced inflammatory macrophages. The results showed that quercetin significantly ameliorated both the morphological alterations (including cell differentiation, spreading and pseudopod formation) and cell proliferation inhibition of macrophages caused by LPS. It is thus known that quercetin protects macrophages in the LPS environment, but its specific action and mechanism remain unclear. To further elucidate the role of quercetin in inflammatory macrophages, we examined the production of ROS, the contents of MDA, GSH and the mRNA expression of pro-inflammatory cytokines in LPS-induced RAW264.7 cells. The results demonstrated that quercetin inhibited ROS production and reduced the mRNA expression of pro-inflammatory cytokines in inflammatory macrophages. These results confirmed that quercetin positively regulates the REDOX balance of inflammatory macrophages and improves the antioxidant capacity of LPS-induced RAW264.7 cells by inhibiting the production of ROS, MDA and the expression of pro-inflammatory cytokines and promoting the production of antioxidant enzymes.

Mitochondria are organelles surrounded by bilayer membranes found in most cells and are the main site of aerobic respiration. Their functions include energy conversion, tricarboxylic acid cycle, oxidative phosphorylation, calcium ion storage, etc. [48]. Mitochondrial stability is critical for the reduction of apoptosis and the promotion of cell growth. Excess ROS generated by mitochondrial oxidative stress leads to dysfunction of the electron transport chain and disrupts the regulation of energy production, ultimately causing mitochondrial damage characterized by reduced mitochondrial membrane potential and disruption of mitochondrial membrane proteins [49,50,51]. Mitochondria-dependent apoptotic pathways are activated when mitochondria are damaged, leading to cellular dysfunction [52]. In order to explore the effect of quercetin on the mitochondria of inflammatory macrophages, transmission electron microscopy was used to observe the morphology of mitochondria in each treatment group. As expected, we found that quercetin mitigated LPS-induced mitochondrial morphological abnormalities in macrophages, suggesting that quercetin has a protective effect on mitochondrial macrophages in an LPS environment. The reduction in mitochondrial membrane potential is generally considered to be a late event in the apoptotic pathway, and it is therefore also regarded as an important biomarker of oxidative-stress-induced apoptosis [53]. The main bioenergetic function of mitochondria is ensured by mitochondrial membrane potential, which decreases with mitochondrial damage and eventually leads to the loss of cellular function [54,55]. JC-1 is one of the common mitochondrial fluorescent probes, it forms aggregate in healthy cells that stain mitochondria red. In contrast, when the mitochondrial membrane potential is reduced, the dye leaks from the mitochondria and appears in a green monomeric form. In this study, JC-1 was used to determine the effect of quercetin on the mitochondrial membrane potential of inflammatory macrophages. The results illustrated that quercetin inhibited the decrease of mitochondrial membrane potential in macrophages caused by LPS.

mtDNA is highly susceptible to oxidative stress due to its intimate relationship with high concentration of ROS, increasing the risk of mitochondrial dysfunction. mtDNA copy number is related to the production of ATP and the activity of mitochondrial enzymes, and is the substitute marker of mitochondrial dysfunction [56,57]. Our study found that there is no significant difference in the mtDNA copy number among different treatment groups, indicating that the protective effect of quercetin on the mitochondria of inflammatory macrophages was not significantly related to the mtDNA copy number. This might be due to the fact that mtDNA copy number changes depend on strict tissue-specific regulation, the mechanism of which is largely unknown [58], and quercetin plays a minor role in this specific regulation, and thus has no significant effect on the downregulation of mtDNA copy number caused by LPS. In addition, the mtDNA copy number is dynamically changing with time [59]. Literature reported that the mtDNA copy number of human embryonic kidney cells (HEK) was reduced to 30% within 48 h under the influence of 2′–3′-dideoxycytidyne (ddC), and returned to the baseline level within 32 h in the absence of ddC [60]. The treatment time of our experiment was 24 h, therefore the experimental results of quercetin’s effect on the mtDNA copy number might be affected by the length of the experimental time and need to be further investigated.

Energy for the maintenance of physiological functions and survival of the organism is mainly supplied by ATP, and this requirement is an unstable process. In addition to their signaling properties, ROS are also degenerative agents that result in aging and disease [61]. Both of them are produced mainly by mitochondria, and their excess determines the fate of cells [62]. According to our experimental results, quercetin has a significantly positive regulatory effect on ATP content in inflammatory macrophages and the opposite effect on ROS. Therefore, quercetin regulates the balance of ATP and ROS in inflammatory macrophages tending to maintain the energy supply of cells. Mitochondrial ATP synthase is a proton-powered ATP generator that uses the mitochondrial electron transport chain and is the main enzymatic complex that produces cellular ATP under aerobic conditions [63,64]. Its defects are directly or indirectly associated with a variety of diseases, including neurodegenerative diseases, retinitis pigmentosa syndromes, cardiomyopathies, etc. [65,66]. It has been reported that the upregulation of ATP synthase activity can alleviate mitochondrial oxidative stress [67]. Consistent with this, our results also demonstrated that quercetin significantly upregulated ATP synthase content in inflammatory macrophages. It could be concluded that the effect of quercetin in alleviating LPS-induced oxidative stress damage may be achieved by protecting the mitochondrial function of inflammatory macrophages. However, in addition to the decrease in ATP synthase activity, the increase in ATP consumption may also be responsible for the decrease in ATP content. Specifically, LPS-induced ROS overproduction led to mitochondrial dysfunction characterized mainly by the disruption of mitochondrial membrane potential. Loss of mitochondrial membrane potential leads to the defective mitochondrial electron transport chain, reduced metabolic oxygen consumption and excessive ATP depletion, and put it into a hypoenergetic metabolic state, thus triggering mitochondrial oxidative stress [68]. In addition, mitochondrial ATP content is also affected by calcium overload [69], hypoxia [70], increased mitochondrial membrane permeability [71] and mitochondrial DNA mutations [72]. The relationship between the effect of quercetin on mitochondrial ATP content and these factors remains to be further studied. Sirtuin 1 (SIRT1) is a conserved nicotinamide adenine dinucleotide (NAD)-dependent mammalian protein deacetylase, with multiple biological functions. It is commonly described to perform critical functions in cell differentiation, senescence, metabolism and apoptosis [73,74,75]. In recent years, an increasing number of studies have reported that SIRT1 can coordinate inflammatory signaling, so it is considered an important target for immune microenvironment regulation. In the study of a murine hepatic ischemia/reperfusion injury model [76], SIRT1 activation alleviated leukocyte infiltration, and higher SIRT1 levels were associated with a lower proinflammatory cytokine profile. In osteolysis models, SIRT1 activation by hydrogen sulfide mitigated the particle-induced inflammatory response and prevented bone resorption [77]. PGC-1α is a multifunctional protein present in the anti-oxidative stress system, which plays a key role in transcriptional regulation by activating most nuclear receptors and co-activating multiple transcription factors [78]. In addition, PGC-1α is an important transcriptional regulator of mitochondrial function, and its activation contributes to increasing the expression of nuclear coding subunits of the mitochondrial respiratory chain [79]. In the presence of oxidative stress, SIRT1 is deacetylated with increasing NAD+/NADH ratio, thereby activating PGC-1α [80]. The increase of PGC-1α regulates cellular response to oxidative stress and induces a significant increase in the gene expression of antioxidant enzymes, including SOD, GPX1, etc. [81]. Notably, the transcriptional activity of PGC-1α is very low when it is not bound to transcription factors, but significantly elevated when it is bound to SIRT1 [82]. Our study found that the protein expression of SIRT1 and PGC-1α was significantly decreased in RAW264.7 cells treated with LPS, while their protein expression was significantly increased by the exogenous use of quercetin. To further demonstrate the role of SIRT1 in the process of quercetin protection of inflammatory macrophages from oxidative stress, we used SIRT1 inhibitor, EX527. Inhibition of SIRT1 significantly attenuated the inhibitory effect of quercetin on LPS-induced ROS production in macrophages, and reversed the protective effect of quercetin on mitochondrial membrane potential and morphological structure of inflammatory macrophages. These results suggest that the protection of inflammatory macrophages from oxidative stress by quercetin is associated with the increased expression of SIRT1. In conclusion, quercetin can ameliorate LPS-induced oxidative damage in inflammatory macrophages via the SIRT1/PGC-1α signaling pathway (Figure 5).

## 4. Materials and Methods

### 4.1. Materials and Reagents

Quercetin and LPS were purchased from Sigma-Aldrich (St. Louis, MO, USA). The mouse macrophage-like cell line, RAW264.7 (CSTR:19375.09.3101MOUTCM13), was acquired from the National Collection of Authenticated Cell Cultures, the Chinese Academy of Sciences (Shanghai, China). Fetal bovine serum (FBS) and Dulbecco’s modified eagle’s medium (DMEM) high glucose were purchased from Gibcol Life Technology (Thermo Fisher, Waltham, MA, USA). An enhanced ATP Assay Kit was acquired from Beyotime^®^ Biotechnology (Shanghai, China). anti-PGC1 (Cat# ab191838), anti-glutathione peroxidase 4 (Cat# ab125066), anti-AMPK alpha 1 (Cat# ab32047) and anti-SIRT1 (Cat# ab110304) antibodies were purchased from Abcam Biotechnology (Cambridge, MA, USA). Phospho-AMPK (Thr172) (Cat# 2535s) antibody was purchased from Cell Signaling Technology (Danvers, MA, USA). The 3-4,5-dimethylthiazole-z-yl-3,5-diphenyltetrazolium bromide (MTT), trypsin, dimethyl sulfoxide (DMSO), phosphate-buffered saline (PBS), LA Assay Kit, Micro Pyruvate (PA) Assay Kit and Mitochondrial Membrane Potential Assay Kit with JC-1 were purchased from Solarbio Science & Technology Co. Ltd. (Beijing, China), and stored at −20 °C. EX-527 and compound C were obtained from Med Chem Express (Monmouth Junction, NJ, USA).

### 4.2. Cell Culture

RAW264.7 macrophages were cultured in DMEM containing 10% FBS at 37 °C, in a fully humidified incubator containing 5% CO_2_. Once grown as a dense monolayer, the cells were routinely passed to a third generation.

### 4.3. Cell Viability Assay

The cell viability was determined by an MTT assay according to a previous procedure, with minor modifications [83]. RAW264.7 cells were seeded in 96-well plates, at a density of 1 × 10^4^ per well in a culture medium. The cells were treated with 0 to 160 µM quercetin for 24 h, then 20 µL of 0.5% MTT was added to each well. Absorbance values were measured at 490 nm after 4 h of MTT addition, by using a spectrophotometer (Epoch Microplate Spectrophotometer, BioTek Instruments, Inc., Winooski, VT, USA). The cell viability was calculated according to the following formula:Cell Viability (%) = [(absorbance of treatment − absorbance of blank)/(absorbance of control − absorbance of blank)] × 100%

### 4.4. Observation of Cell Morphological Changes

RAW264.7 cells were plated in 6-well plates with a density of 6 × 10^5^ per well, and cultured for 24 h. Subsequently, the blank control group was treated with PBS for 24 h; the quercetin group was treated with quercetin for 24 h; the LPS group was treated with PBS for 12 h at first, followed by adding LPS (1 µg/mL) for another 12 h stimulation; the quercetin-treated group was treated with quercetin alone in the first 12 h and then LPS (1 µg/mL) was added with quercetin for co-treatment in the second 12 h. The morphological changes of the cells in each group were observed by the LASEZ Microscope Assisted Imaging System.

### 4.5. Cell Proliferation Assay

The changes of cell proliferation were determined using a previously described procedure [84], with some modifications. RAW264.7 cells were treated as described in Section 2.4. EdU staining was conducted using the BeyoClick™ EdU Cell Proliferation Kit with Alexa Fluor 594 (Beyotime, Shanghai, China, Cat. No. C00788L). The treated cells were washed with PBS and then fresh DMEW was added, containing 10 μM EdU. After incubation at 37 °C for 2 h, the cells were washed again with PBS to remove DMEM and free EdU probes. Cells were then fixed with 4% paraformaldehyde for 30 min at room temperature, followed by DAPI staining for 3 min. After an additional wash in PBS, the cells were observed under a laser-scanning confocal microscope (Model: ZEISS LSM800). The intensities of fluorescence were analyzed by the ImageJ software (version No. 1.53e). The percentage of EdU-positive cells was calculated according to the following formula:EdU-positive cell (%) = EdU (Red fluorescence)/Hoechst (Blue fluorescence) × 100%

### 4.6. Determination of Cellular ROS Production

The ROS production in RAW264.7 cells was determined by dichloro-dihydro-fluorescein diacetate (DCFH-DA) assay according to an earlier reported method [85], with slight modifications. Briefly, cells inoculated on the confocal dish were cultured at 37 °C for 24 h. The confocal dish was washed with PBS (100 μL), and the growth medium was later removed. Then, different concentrations of PCA were added to pre-protect for 30 min, followed by 12 h of treatment with quercetin (10 uM). Finally, the cells were incubated with DCFH-DA (Jiancheng Bioengineering Institute, Nanjing, China), with a final concentration of 10 μM for 30 min at 37 °C, and then washed twice with PBS. DCFH-DA was hydrolyzed to DCFH carboxylate anion by cellular esterases and then oxidized by ROS to highly fluorescent dichlorofluorescein (DCF). Fluorescence was observed by confocal microscopy.

### 4.7. Evaluation of Enzyme Activity

The levels of oxidative stress in cell samples were assessed by measuring enzyme activity. RAW264.7 cells were treated as described in Section 2.4. The cells of each group were lysed and their concentrations of total protein were measured with the BCA protein assay kit (TaKaRa Bio, Shiga, Japan), respectively. The activity of reduced glutathione (GSH) and the content of malondialdehyde (MDA) in cells were determined by using commercial kits, according to the manufacturer’s instructions (Nanjing Jiancheng Technology Co., Ltd., Nanjing, China).

### 4.8. Detection of mRNA Expression

The mRNA expression levels of inflammatory factors (IL-1β, TNF-α, IL-6 and NF-κB) were detected by quantitative real-time polymerase chain reaction (qRT-PCR). RAW264.7 cells were treated as described in Section 2.4. Total RNA of each group of cells was extracted using the TRIzol reagent (Vazyme, Nanjing, China), and reverse transcribed into cDNA using the Prime Script RT reagent kit (Thermo Fisher, Waltham, MA, USA). Quantitative PCR was performed using ChamQ Universal SYBR qPCR Master Mix (Vazyme, Nanjing, China). The reaction was performed at a total volume of 10 μL, with the assay solution containing 5 μL ChamQ Universal SYBR qPCR Master Mix, 3.6 μL deionized H_2_O, 1 μL cDNA template, and 0.2 μL each of the forward and reverse primers. The fold changes in mRNA expression were calculated by comparing the β-actin normalized threshold cycle numbers (Ct), using the 2^−ΔΔCT^ method. Triplicate wells were run for each experiment and two independent experiments were performed. The primer sequences designed for qRT-PCR analysis are listed in Table 1.

### 4.9. Detection of Effects on Mitochondria

#### 4.9.1. Detection of Mitochondrial Membrane Potential

The changes in mitochondrial membrane potential (Δψm) were observed with JC-1 staining dye assay kit (Beijing Solarbio Science & Technology Co., Ltd., Beijing, China), according to the method described by early literature [86]. JC-1 dye is a lipophilic cationic fluorescent dye with dual (red and green) emission wavelengths, and the attenuation of its red/green fluorescence intensity ratio is commonly used to represent an increase in mitochondrial membrane depolarization. The cells were cultured on the confocal dish with different treatments. Then, the old medium was replaced with fresh medium containing JC-1. Cells continued to incubate at 37 °C for 25 min and were then washed twice with ice-cold PBS. Cell morphology and staining were observed and photographed by an inverted fluorescence microscope (ZEISS Axio Observer A1, Oberkochen, German) and the intensities of fluorescence were analyzed by the ImageJ software (version No. 1.53e).

#### 4.9.2. Observation of Mitochondrial Morphology

The morphological changes of mitochondria were observed by transmission electron microscopy (TEM). RAW264.7 cells were treated as described in Section 2.4. The medium of the treated cells in each group was discarded and 3% glutaraldehyde was rapidly added. The cells were gently scraped and collected into centrifuge tubes, then centrifuged at 1000 rpm for 5 min after discarding glutaraldehyde. Glutaraldehyde (3%) was added again to fix the cells, followed by graded alcohol dehydration, resin embedding, and ultrathin section staining with uranyl acetate and citric acid. Autophagosomes and cellular mitochondria were observed by transmission electron microscopy (JEM-1400 Flash, Tokyo, Japan).

#### 4.9.3. Determination of Mitochondrial DNA Copy Number

RAW264.7 cells were treated as described in Section 2.4. After treatments, total DNA of the cells in each group was extracted by the Animal Tissues/Cells Genomic DNA Extraction Kit (Beijing Solarbio Science & Technology Co., Ltd., Beijing, China), according to the manufacturer’s instructions. The concentrations of extracted DNA were quantified by measuring absorbance at 260 nm, and then quantitative PCR of DNA was performed using ChamQ Universal SYBR qPCR Master Mix (Vazyme, Nanjing, China). Mitochondrial DNA copy numbers were assessed by using primers targeting mitochondria-encoded mtDNA genes (forward: 5′-ATCCTCCCAGGATTTGGAAT-3′; reverse-5′ACCGGTAGGAATTGCGATAA-3′). Primers designed against the nuclear-encoded 18s RNA gene (forward: 5′-TTCGGAACTGAGGCCATGATT-3′; reverse: 5′-TTTCGCTCTGGTCCGTCTTG-3′) were used for normalization. The mitochondrial DNA copy number was calculated by using the 2^−ΔΔCT^ method.

#### 4.9.4. Determination of Mitochondrial Function

##### ATP Contents

RAW264.7 cells were treated as described in Section 4.4. Lysis solution (200 µL) was added to each well after removing the medium, the cells were lysed and then collected into centrifuge tubes. The lysed cells were centrifuged at 4 °C with 12,000 rpm for 5 min. The supernatant was collected and the ATP content was measured by the enhanced ATP assay kit, according to the manufacturer’s instructions (Beyotime^®^ Biotechnology, Shanghai, China).

##### ATP Synthase Content

RAW264.7 cells were treated as described in Section 4.4. Briefly, the treated cells were collected and lysed, and the supernatant was collected by centrifugation. The content of ATP synthase was determined by the ATP synthase assay kit, according to the manufacturer’s instructions (Jiangsu Meimian Industrial Co., Ltd., Yancheng, China).

### 4.10. Western Blotting Analysis

RAW264.7 cells were treated as described in Section 2.4. After treatment, the culture medium of each group was discarded, and RIPA lysis buffer, containing PMSF, was added. The cells were scraped gently and collected into centrifuge tubes. After centrifugation at 4 °C with 12,000 rpm for 30 min, the supernatants were collected and their concentrations of total protein were measured by BCA protein assay kit. The obtained samples were mixed with 5 × loading buffer and heated in boiling water for 10 min to denature proteins, then resolved in 10% SDS-PAGE gels and transferred to NC membranes. The membranes were blocked with QuickBlock™ Blocking Buffer for 15 min after TBST buffer washing, and then incubated for 12–16 h at 4 °C with corresponding antibody solutions (1:1000). After washing, the membranes were incubated with secondary antibodies for 1 h at 37 °C. The chemiluminescence-positive signals were detected by the ECL Western blotting detection reagent (Cat. No. 34079, Thermo Scientific, Waltham, MA, USA), the protein band images were scanned and analyzed as the integrated absorbance (IA = mean OD × area) using the ImageJ software (version No. 1.53e), and the relative levels of target proteins were normalized to β-actin (target protein IA/β-actin IA).

### 4.11. SIRT-1 Inhibition Assay

In this section, the EX-527 group was added in addition to the four cell treatment groups described in Section 2.4. The EX-527 group was treated with quercetin for 12 h, and then LPS (1 µg/mL) and EX-527 (10 μM) were added to co-treat for 12 h. The ROS expression content, mitochondrial membrane potential and mitochondrial morphological changes of the cells in each treatment group were detected and observed according to methods described in Section 4.6, Section 4.9.1 and Section 4.9.2, respectively.

### 4.12. Statistical Analysis

Statistical analyses were performed by the GraphPad Prism software (version No. 9.0; GraphPad Software, Inc.). The data of experiments were expressed as the mean ± standard deviation of three independent experiments, and comparisons between groups were first tested for normal distribution using the Shapiro–Wilk test, then parametric testing using the Brown–Forsythe test, and finally, a one-way ANOVA with Tukey’s post hoc test. *p* < 0.05 was considered statistically significant.

## 5. Conclusions

In summary, the present study revealed that quercetin alleviates LPS-induced oxidative stress damage in macrophages by promoting antioxidant enzyme activity, inhibiting ROS production and inflammatory factor overexpression, and protecting mitochondrial function. Furthermore, SIRT1 and PGC-1α have been shown to play important roles in the reprogramming of inflammatory macrophage metabolism by quercetin. The alleviating effect of quercetin on LPS-induced oxidative stress damage in macrophages is exerted through the SIRT1/PGC-1α signaling pathway, which participates in the mitochondrial metabolism reprogramming. However, the discussion of quercetin’s oxidative stress alleviating effect in this study is limited to in vitro experiments, the in vivo mechanism verification experiments will be conducted in the next research work. In addition, this study will further explore the effect of quercetin treatment on mitochondrial electron respiratory chain under oxidative stress conditions, from the perspective of cell energy metabolism, such as mitochondrial respiratory chain complex I, III, etc., in the hope of laying a theoretical foundation for the application of quercetin in the prevention and treatment of oxidative-stress-related diseases.

## Figures and Tables

**Figure 1 ijms-24-05542-f001:**
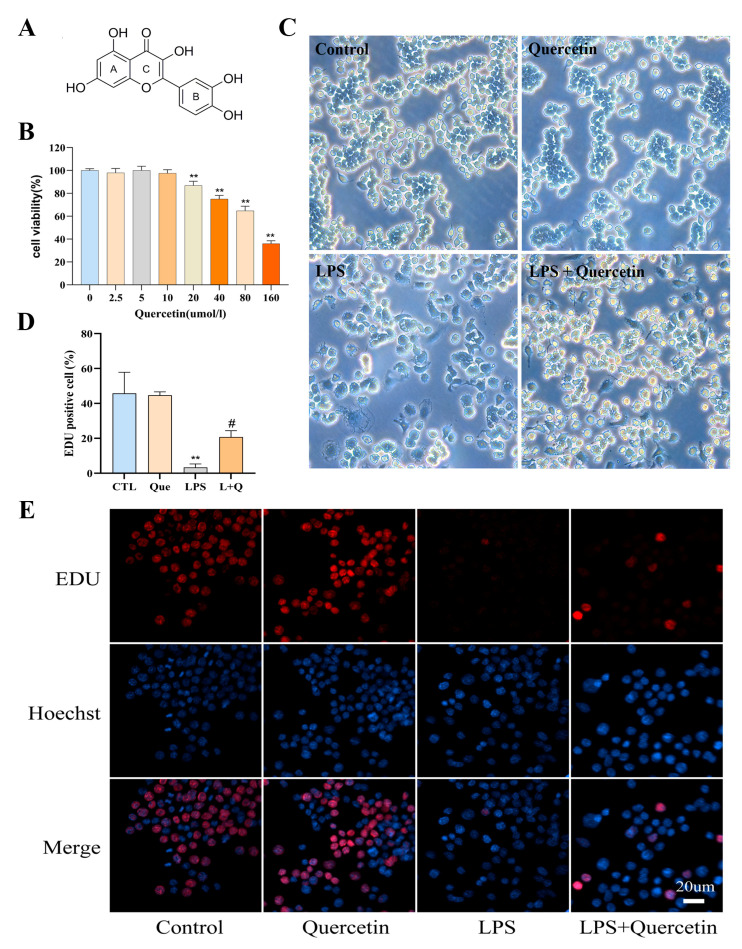
Effect on quercetin on RAW264.7 cells. (**A**) Chemical structure of quercetin. (**B**) The effect of quercetin on the viability of RAW264.7 cells. (**C**) The effect of quercetin on morphology of LPS-induced RAW264.7 cells. (**D**) Percentage of EdU positive cells. (**E**) The effect of quercetin in the proliferation of LPS-induced RAW264.7 cells. ** *p* < 0.01 compared to the blank control group, and ^#^ *p* < 0.05 compared to the LPS group were considered statistically significant differences.

**Figure 2 ijms-24-05542-f002:**
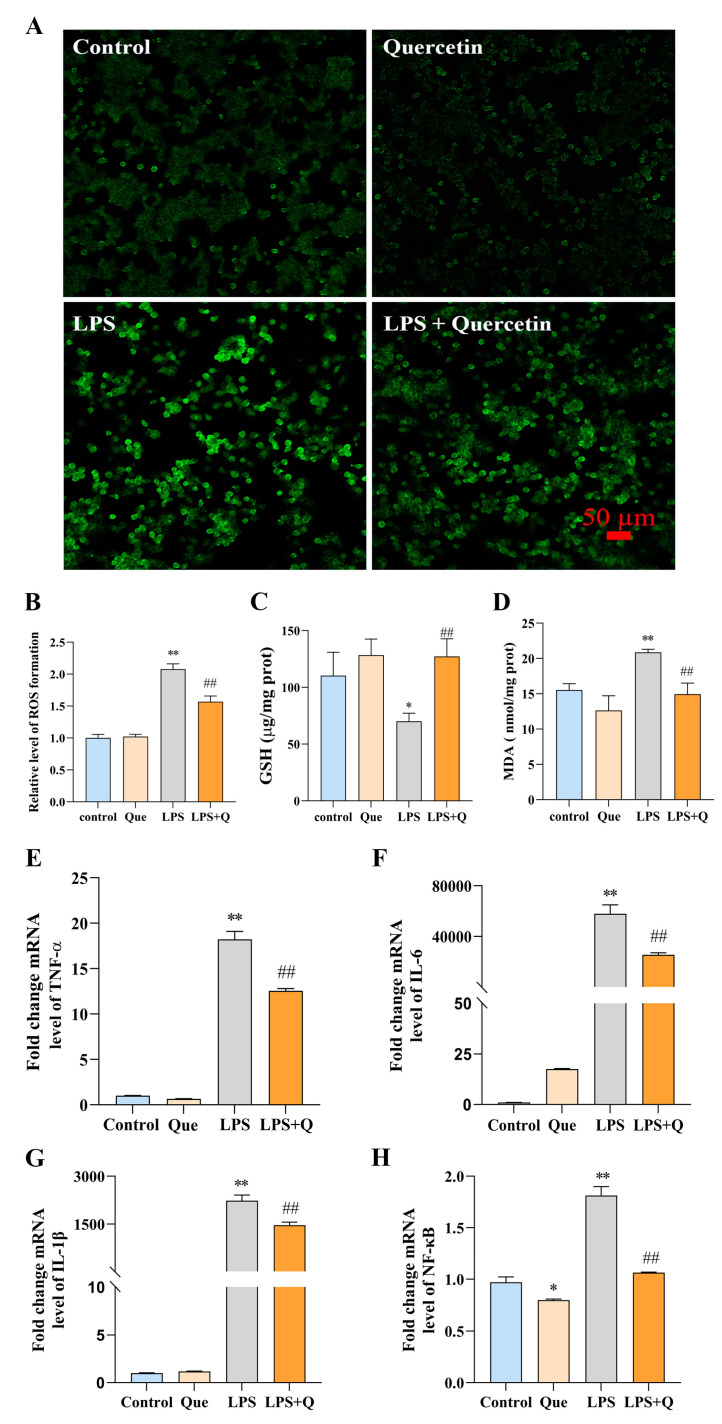
Effects of quercetin on antioxidant capacity and expression of inflammatory cytokines in LPS-induced RAW267.4 cells. (**A**) Images of DCF fluorescence intensity. (**B**) The relative levels of ROS generation. (**C**,**D**) The effect of quercetin on the intracellular levels of GSH and MDA. The expression of (**E**) TNF-α, (**F**) IL-6, (**G**) IL-1β and (**H**) NF-κB mRNA levels were determined by qRT-PCR. * *p* < 0.05 and ** *p* < 0.01 compared to the blank control group, and ^##^ *p* < 0.01 compared to the LPS group were considered statistically significant differences.

**Figure 3 ijms-24-05542-f003:**
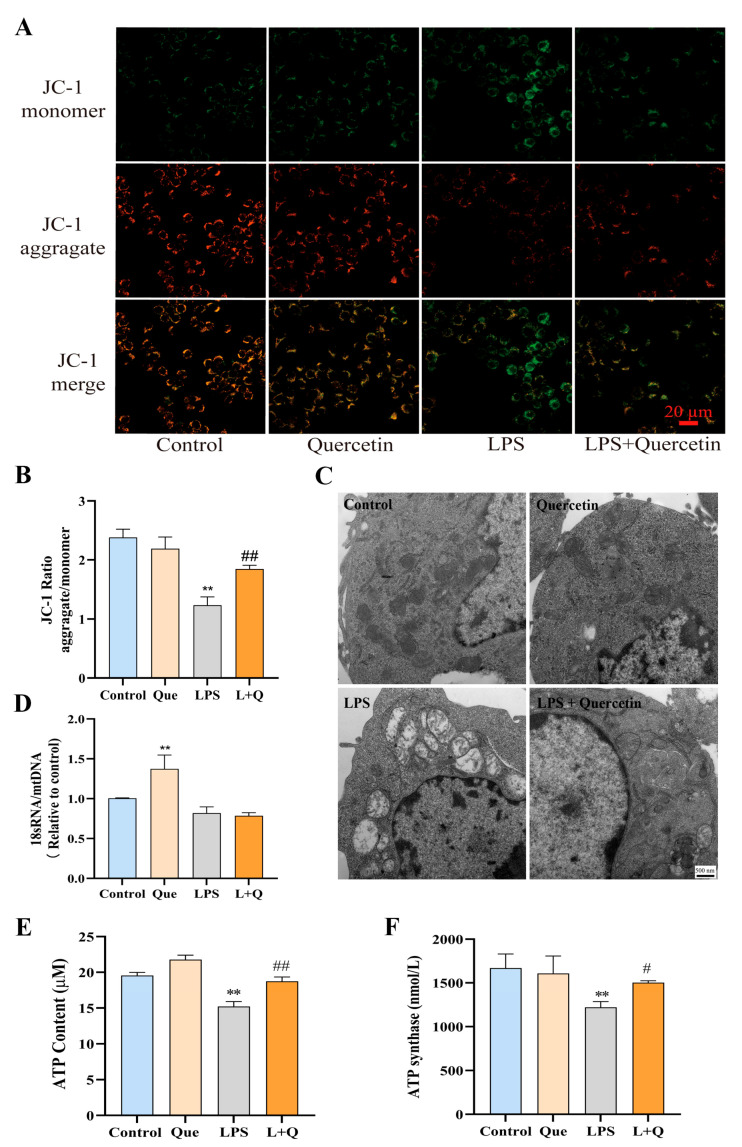
Effects of quercetin on the mitochondria of LPS-induced RAW267.4 cells. (**A**) Fluorescence intensity of cells stained with JC-1. Fluorescence intensity was measured using a multimode microplate reader at 485 nm excitation, 585 (red/orange for normal Δψ) and 538 (green for loss of Δψ) emissions, respectively. (**B**) The red/green fluorescence intensity ratio of JC-1 dye. (**C**) Transmission electron microscope images of mitochondria. (**D**) The mtDNA copy number of each treatment group. (**E**,**F**) The contents of ATP and ATP synthetase in each treatment group. ** *p* < 0.01 compared to the blank control group, and ^#^ *p* < 0.05 and ^##^ *p* < 0.01 compared to the LPS group were considered statistically significant differences.

**Figure 4 ijms-24-05542-f004:**
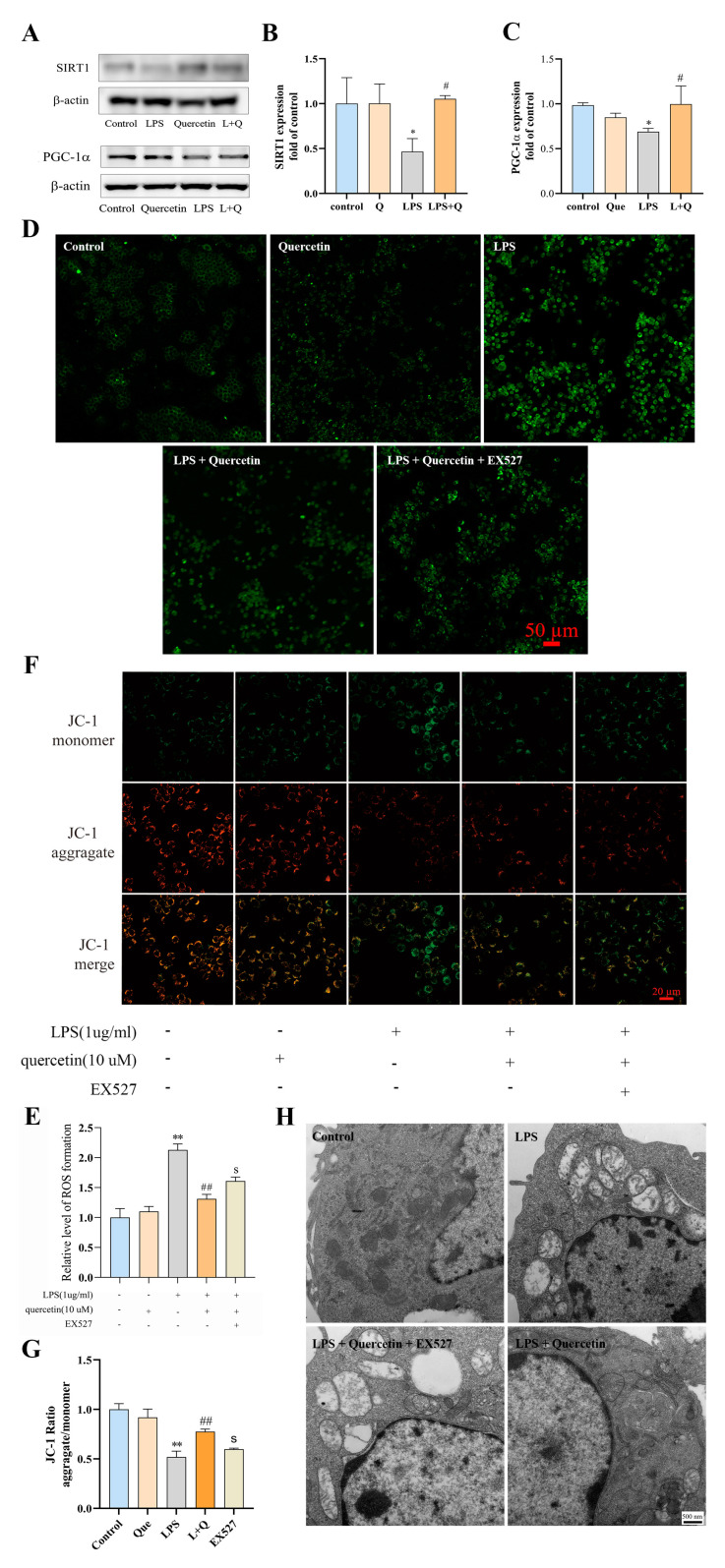
(**A**–**C**) Representative protein bands and the histograms of band intensity analysis of SIRT1 and PGC-1a in LPS-induced RAW264.7 cells. (**D**,**E**) Images of DCF fluorescence intensity and histogram of ROS production in each group. (**F**,**G**) Images of JC-1 fluorescence intensity and histogram of JC-1 ratio in each group. (**H**) Comparison of transmission electron microscopy images of mitochondria. * *p* < 0.05 and ** *p* < 0.01 compared to the blank control group, ^#^ *p* < 0.05 and ^##^ *p* < 0.01 compared to the LPS group, and ^S^
*p* < 0.05 compared to the quercetin-treated group were considered statistically significant differences.

**Figure 5 ijms-24-05542-f005:**
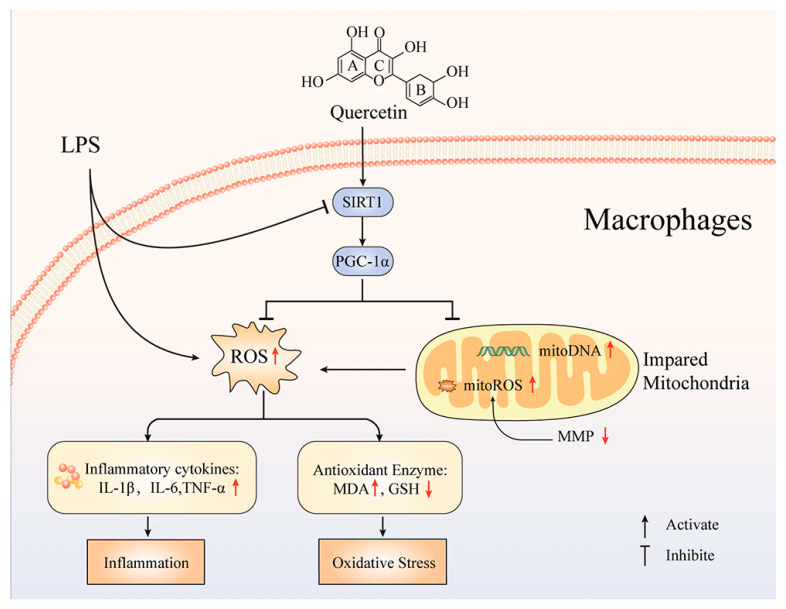
The process of quercetin mitigating LPS-induced oxidative damage in inflammatory macrophages. The upward red arrow indicates positive regulation, while the downward red arrow indicates negative regulation.

**Table 1 ijms-24-05542-t001:** Primers used for qRT-PCR.

Gene Symbol	Sequence (5′ → 3′)
IL-1β	(F) TCGCTCAGGGTCACAAGAAA
	(R) CATCAGAGGCAAGGAGGAAAAC
IL-6	(F) TCTATACCACTTCACAAGTCGGA
	(R) GAATTGCCATTGCACAACTCTTT
TNF-α	(F) GCCACCACGCTCTTCTGTCT
	(R) GTCTGGGCCATAGAACTGAT
NF-κB	(F) GAGGTCTCTGGGGGTACCAT
	(R) TTGCGGAAGGATGTCTCCAC
β-actin	(F) GAAGATCAAGATCATTGCTCCT
	(R) TGGAAGGTGGACAGTGAG

## Data Availability

The original contributions presented in the study are included in the article. Further inquiries can be directed to the corresponding authors.

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
