# Peer review of "Quercetin Reprograms Immunometabolism of Macrophages via the SIRT1/PGC-1α Signaling Pathway to Ameliorate Lipopolysaccharide-Induced Oxidative Damage"

_ijms, 2023, doi:10.3390/ijms24065542_

Round 1

Reviewer 1 Report

The paper is interesting and the conclusions appear generally in agreement with experimental data, moreover the Authors should meliorate the article:

1)     in the text at 155, 156 and 157 lines please correct the unit of measure mg with mM;

2)     since  474 line you indicated the exposure time of quercetin 24 h; LPS PBS 12 h followed LPS 12 h; instead quercitin treated group: 12 h quercitine then LPS was add to co-treat for 12h; please explain better: It could appear that you added quercitin before LPS. Perhaps this aspect have to be clarified and the time of this co-treat is not well clear; you might add a figure.   

3)     since 258 line you affirm that mtDNA can reflect mitochondrial damage deriving from oxidative stress, but your results are negative; in the discussion you should explain this result that does not appear in agreement with other data showing an important effect on mitochondrial  function;

4)     ATP content might depend on higher consuption also if ATP synthase decreased? Please discuss this eventuality explaining as quercitin increased it   

Author Response

Response to Reviewer 1:

Dear Reviewer,

Thank you very much for your careful and comprehensive review of our paper. Based on your comment and suggestion, we revised the relevant part in manuscript. The responses answering every question are as follows. All the lines and pages mentioned in responses are form the revised manuscript in review mode (Word version).

Comments and Suggestions. The paper is interesting and the conclusions appear generally in agreement with experimental data, moreover the Authors should meliorate the article.

Response: Thank you for your comments and suggestions. We have read your comments carefully. Thank you again for your careful and rigorous review of the article and your affirmation of our work. We will carefully revise the article according to the comments and suggestions you listed. Thank you!

Comment 1. In the text at 155, 156 and 157 lines please correct the unit of measure mg with mM.

Response: Thank you for your suggestion. We are sorry for this inattention. We have revised the incorrect unit in the manuscript (page 4, lines 155-157). Thank you.

Comment 2. Since 474 line you indicated the exposure time of quercetin 24 h; LPS PBS 12 h followed LPS 12 h; instead quercetin treated group: 12 h quercetin then LPS was add to co-treat for 12h; please explain better: It could appear that you added quercetin before LPS. Perhaps this aspect have to be clarified and the time of this co-treat is not well clear; you might add a figure.

Response: Thank you for your comment and suggestion. We apologize for the unclear description. The total treatment time for all experimental groups was 24 h. The blank control group and quercetin group were treated with PBS and quercetin alone for 24 h, respectively. LPS group was treated with PBS for 12 h at first, followed by adding with LPS (1 µg/mL) for another 12 h treatment. Quercetin-treated group was treated with quercetin alone in the first 12 h and then LPS (1 µg/mL) was added with quercetin for co-treatment in the second 12 h. The treatment method of each group was shown in the figure below. We have revised relevant part in the manuscript (page 14, lines 474-477). Thank you.

Comment 3. Since 258 line you affirm that mtDNA can reflect mitochondrial damage deriving from oxidative stress, but your results are negative; in the discussion you should explain this result that does not appear in agreement with other data showing an important effect on mitochondrial function.

Response: Thank you for your deep comment and suggestion. We have revised relevant part in the manuscript (page 12, lines 384-394). Thank you.

Comment 4. ATP content might depend on higher consumption also if ATP synthase decreased? Please discuss this eventuality explaining as quercetin increased it.

Response: Thank you for your deep comment and suggestion. We have revised relevant part in the manuscript (page 19, lines 447-460). Thank you.

Reviewer 2 Report

Generally, the topic is interesting and fits into the scope of the journal. However, some parts of the manuscript should be improved, some of them should be added. For example, the abstract should mainly contain the description of your own results, not general statements and the Results section should be more quantitative, i.e., it would be welcome to provide % of changes observed. Further, statistics is very important and this point should be clarified in detail. My most serious concerns focus on the methods of the statistical analysis. The kind of test used to reveal the statistically significant differences among groups should be provided. Moreover, was the normal distribution test performed for statistical analysis, how was the parametric test decision made? There is also no information referring to the homogeneity of variances - whether the homogeneity of variances was verified. Moreover, the Authors should explain why did they choose these particular doses of quercetin. In addition, the conclusion section should be provided - it seems to be missing now. I also strongly recommend to include future perspective of the study, i.e., what is the next step? Besides, what are the limitations of the study?

Furthermore, the points listed below need to be checked by the Authors.

1.     Line 170: Figure 1. ‘Effect on….’

2.     NF-kB and NF-kb, please unify

3.     Line 274: Wouldn’t it be better to describe ‘treatment with quercetin alone’ instead of ‘quercetin treatment alone…’?, please check throughout the manuscript.

4.     Line 275: Please consider another word instead of ‘treatment’ when you use LPS

5.     Line 510: glutathione peroxidase (GSH), the reviewer think that the Authors meant GPx

6.     Line 604: ‘extremaly statistically significat’. What does it exactly mean?

7.     Figure 1C and D, Figure 2A, 3A and C, and Figure 4 D, F, H need to be improved in terms of quality.

Author Response

Dear Reviewer,

We are very grateful to your comments for the manuscript. According with your comments and suggestions, we amended the relevant part in manuscript. All of the comments and suggestions are responded as follows. All the lines and pages mentioned in responses are form the revised manuscript in review mode (Word version).

Comments and Suggestions. Generally, the topic is interesting and fits into the scope of the journal. However, some parts of the manuscript should be improved, some of them should be added.

Response: Thank you for your pertinent comments and constructive suggestions. Your affirmation gives us great encouragement. We have carefully revised the manuscript based on each comment and suggestion you pointed. Thank you.

Comment 1. The abstract should mainly contain the description of your own results, not general statements.

Response: Thank you for pointing this out. We have rewritten the abstract in the manuscript (page 1, lines 12-39) according your advice. Thank you.

Comment 2. The Results section should be more quantitative, i.e., it would be welcome to provide % of changes observed.

Response: Thank you for your constructive comments. We have quantified the results section based on your suggestions and revised relevant parts in the manuscript. Thank you.

Comment 3. Statistics is very important and this point should be clarified in detail. My most serious concerns focus on the methods of the statistical analysis. The kind of test used to reveal the statistically significant differences among groups should be provided. Moreover, was the normal distribution test performed for statistical analysis, how was the parametric test decision made? There is also no information referring to the homogeneity of variances - whether the homogeneity of variances was verified.

Response: Thank you for your deep comment and suggestion. First of all, we would like to apologize for wrongly writing "SPSS software" instead of "GraphPad Prism software" in the original manuscript. The comparisons between groups were first tested for normal distribution using the Shapiro-Wilk test, then parametric testing using the Brown-Forsythe test, and finally one-way ANOVA using Tukey's post hoc test. We have revised Statistical analysis section in the manuscript (page 24, lines 669-675). In addition, the statistical analysis data for each experiment was compiled and uploaded as supplementary data. Thank you.

Comment 4. Moreover, the Authors should explain why did they choose these particular doses of quercetin.

Response: Thank you for your comment and suggestion. We are sorry for this carelessness. We have revised relevant part in the manuscript (page 4, lines 175-180). Thank you.

Comment 5. In addition, the conclusion section should be provided - it seems to be missing now.

Response: Thank you for pointing this out. We are sorry for our negligence. We have added the conclusion section in the manuscript (page 24, lines 676-691). Thank you.

Comment 6. I also strongly recommend to include future perspective of the study, i.e., what is the next step? Besides, what are the limitations of the study?

Response: Thank you for your insightful comments and practical suggestion. We have included the description of the limitations and future prospects of this study in the manuscript (page 18, lines 420-430; page 19, lines 447-460; page 24, lines 684-691). Thank you.

Comment 7. Line 170: Figure 1. ‘Effect on….’.

Response: Thank you for pointing this out. We are sorry for the carelessness. We have revised it in the manuscript (page 6, line 192). Thank you.

Comment 8. NF-kB and NF-kb, please unify.

Response: Thank you for pointing this out. We are sorry for our negligence. We have carefully reviewed and corrected the relevant errors in the manuscript. Thank you.

Comment 9. Line 274: Wouldn’t it be better to describe ‘treatment with quercetin alone’ instead of ‘quercetin treatment alone…’? please check throughout the manuscript.

Response: Thank you for your helpful suggestions. We have checked throughout the manuscript and revised all of the relevant descriptions. Thank you.

Comment 10. Line 275: Please consider another word instead of ‘treatment’ when you use LPS.

Response: Thank you for your comment and suggestion. We have revised the inappropriate word "treatment" with "stimulation" in the manuscript. Thank you. 

Comment 11. Line 510: glutathione peroxidase (GSH), the reviewer think that the Authors meant GPx.

Response: Thank you for pointing this out. In this study, we actually measured the amount of reduced glutathione (GSH). “Reduced glutathione” was incorrectly written by us in the manuscript as “glutathione peroxidase”. We apologize for any misunderstanding this error may have caused you. We have revised it in the manuscript (page 22, line 578). Thank you.

Comment 12. Line 604: ‘extremaly statistically significat’. What does it exactly mean?

Response: Thank you for your comment and suggestion. Our intention in using "extremely statistically significant" was to distinguish the difference in the degree of significance between "P < 0.05" and "P < 0.01". This might be an inappropriate expression, and we are sorry for any misunderstanding this error may have caused you. We have revised it in the manuscript (page 24, line 675). Thank you.

Comment 13. Figure 1C and D, Figure 2A, 3A and C, and Figure 4 D, F, H need to be improved in terms of quality.

Response: Thank you for your comment and suggestions. We have redrawn Figures 1-4 and improved the quality of the Figures you mentioned. Thank you.

Reviewer 3 Report

1. In this manuscript, authors thought that quercetin inhibited oxidative damage,

  and found that quercetin could decrease MDA and promote GSH levels in

LPS-induced cells. The authors did not investigate the molecular mechanism by

which quercetin mitigates oxidative damage.

2.Figure 2A, Confocal microscope images of the fluorescence intensity. However,

  this figure 2A seems to be a fluorescent microscope image, not confocal

microscope

3. Figure 5. quercetin mitigated mitoROS, and detected by DCF fluorescence.

However, MitoSOX reagent or other mitochondria specific reagent should be more suitable than DCFDA.

Author Response

Response to Reviewer 3:

Dear Reviewer,

Thank you very much for your comments and suggestion concerning our manuscript. The comments and suggestions are valuable and very helpful for revising and improving our paper, as well as the important guiding significance to our researches. We have taken all these comments and suggestions into account, and have made major corrections in this revised manuscript. The responses answering every question are as follows. All the lines and pages mentioned in responses are form the revised manuscript in review mode (Word version).

Comment 1. In this manuscript, authors thought that quercetin inhibited oxidative damage, and found that quercetin could decrease MDA and promote GSH levels in LPS-induced cells. The authors did not investigate the molecular mechanism by which quercetin mitigates oxidative damage.

Response: Thank you for your comment and suggestion. We apologize for any confusion our weak presentation ability may have caused you. The abstract of the original manuscript failed to properly summarize our research results, so we first re-enforced the writing of the abstract section (page 1, lines 12-39). Secondly, our experiments were designed in three parts (as shown in the figure below). Part I: The effects of quercetin on LPS-induced macrophage oxidative damage were investigated by establishing the model of LPS-induced oxidative damage in macrophages. The experiments included the detection of cell activity and proliferation, intracellular ROS production, and the activity of intracellular anti-inflammatory factors and antioxidant enzymes. The results showed that quercetin attenuated the effect of LPS on macrophage proliferation, inhibited LPS-induced cell spreading and the formation of pseudopodia, increased the antioxidant enzyme activity of inflammatory macrophages and inhibited their ROS production and inflammatory factor overexpression, thereby alleviated LPS-induced oxidative damage in macrophages. Part II: In view of the important role of mitochondria in the growth and apoptosis of macrophages (Rambold et al., 2018; Geeraerts et al., 2017), we investigated whether the alleviating effect of quercetin on oxidative damage in macrophages was related to its effect on mitochondria based on the established model of oxidative damage in macrophages. The experiments include the observation of mitochondrial morphology and the detection of mitochondrial function. The results demonstrated that quercetin reversed the mitochondrial morphological damage caused by LPS to some extent, and up-regulated the LPS-induced decrease in mitochondrial membrane potential, ATP production and ATP synthase content. It was speculated that quercetin alleviates LPS-induced oxidative damage to macrophages by restoring impaired mitochondrial morphology and function. Part III: SIRT1 is an important metabolic sensor that coordinates changes in energy metabolism, and its activation promotes oxidative metabolism and coordinates inflammatory signals (Schug et al., 2010; Wang et al., 2011). PGC-1α is a major target of SIRT1, coordinating transcriptional changes in response to SIRT1 activity (Rodgers et al., 2005). Deacetylation of SIRT1 enhances the activity of PGC-1α and induces the transcription of its target genes, thus promoting mitochondrial biogenesis and oxidative metabolism (Rodgers et al., 2005; Nemoto et al., 2005). Therefore, we investigated the role of SIRT1 and PGC-1α in the regulation of mitochondrial function and mitigation of LPS-induced oxidative damage in macrophages by quercetin. The results illustrated that quercetin significantly up-regulated the protein expressions of SIRT1 and PGC-1α, that were inhibited by LPS. And the inhibitory effects of quercetin on LPS-induced ROS production in macrophages and the protective effects on mitochondrial morphology and membrane potential were significantly decreased by the addition of SIRT1 inhibitors. By combining the experimental results of the three parts, we concluded the inference of this paper that quercetin reprograms the immune metabolism of macrophages through the SIRT1/PGC-1α signaling pathway, thereby alleviating LPS-induced oxidative stress damage. However, there are still limitations and a lot of improvement in our study. For example, the discussion on the effect of quercetin on relieving oxidative stress in this study was limited to in vitro experiments, and further in vivo mechanism verification experiments will be conducted in the following research work. In addition, the future of this study hopes to explore the effect of quercetin treatment on mitochondrial electron respiratory chain under oxidative stress condition from the perspective of cell energy metabolism, so as to more in-depth investigate the molecular mechanism of quercetin in mitigating oxidative damage. Thank you again for your valuable comments on the shortcomings of our work, which will urge us to make continuous progress. Thank you.

Figure 1. Graphic abstract of experimental design

1. Rambold, A. S., Pearce, E. L. (2018). Mitochondrial dynamics at the interface of immune cell metabolism and function. Trends Immunol. 39 (1), 6-18. doi: 10.1016/j.it.2017.08.006

2. Geeraerts, X., Bolli, E., Fendt, S. M., Ginderachter, J. A. (2017). Macrophage metabolism as therapeutic target for cancer, atherosclerosis, and obesity. Front Immunol. 8, 289. doi: 10.3389/fimmu.2017.00289

3. Schug, T. T., Xu, Q., Gao, H., Peres-da-Silva, A., Draper, D.W., Fessler, M. B, et al. (2010). Myeloid deletion of SIRT1 induces inflammatory signaling in response to environmental stress. Mol Cell Biol. 30, 4712-4721. doi: 10.1128/MCB.00657-10

4. Wang, R. H., Kim, H. S., Xiao, C., Xu, X., Gavrilova, O., Deng, C. X. (2011). Hepatic Sirt1deficiency in mice impairs mTorc2/Akt signaling and results in hyperglycemia, oxidative damage, and insulin resistance. J Clin Invest. 121, 4477-4490. doi: 10.1172/JCI46243

5. Rodgers, J. T., Lerin, C., Haas, W., Gygi, S. P., Spiegelman, B. M., Puigserver, P. (2005). Nutrient control of glucose homeostasis through a complex of PGC-1a and SIRT1. Nature. 434, 113-118. doi: 10.1038/nature03354

6. Nemoto, S., Fergusson, M. M., Finkel, T. (2005). SIRT1 functionally interacts with the metabolic regulator and transcriptional coactivator PGC-1a. J Biol Chem. 280, 16456-16460. doi: 10.1074/jbc.M501485200

Comment 2. Figure 2A, Confocal microscope images of the fluorescence intensity. However, this figure 2A seems to be a fluorescent microscope image, not confocal microscope.

Response: Thank you for pointing this out. We are sorry for the inappropriate description. The image in Figure 2A was collected with Zeiss LSM 800 confocal laser scanning microscope. However, as you suggested, the description of it should be changed. We have revised it in the manuscript (page 9, lines 219-220). Thank you.

Comment 3. Figure 5. quercetin mitigated mitoROS, and detected by DCF fluorescence. However, MitoSOX reagent or other mitochondria specific reagent should be more suitable than DCFDA.

Response: Thank you for your comment and suggestion. The plot of Figure 5 was based on the combination of published literatures and our experimental results. It has been reported that the decrease of mitochondrial membrane potential leads to the disorder of mitochondrial electron transport chain, which causes the increase of mitochondrial ROS (Fock et al., 2021). In our research, we found that quercetin could restore the mitochondrial morphological damage caused by LPS to some extent and inhibit the decrease of mitochondrial membrane potential caused by LPS by observation of mitochondrial morphology and detection of mitochondrial function. Meanwhile, by investigating the molecular mechanism of quercetin alleviating oxidative damage in macrophages, it was found that quercetin significantly upregulate the protein expression of SIRT1 and PGC-1a inhibited by LPS, while the inhibitory effect of quercetin on LPS-induced ROS production in macrophages, the protective effect on mitochondrial morphology and the upregulation effect of mitochondrial membrane potential were significantly decreased after the addition of SIRT1 inhibitor. Therefore, we described in Figure 5 that quercetin inhibits the membrane potential decline of damaged mitochondria by activating the SIRT1/PGC-1a signaling pathway, thereby affecting the production of mitochondrial ROS. The ROS production we examined in the article were all intracellular ROS and therefore DCF-DA was used. We will explore the changes of the mitochondrial electron respiratory chain under oxidative stress condition after quercetin treatment from the perspective of cellular energy metabolism even further in the next studies and use mitochondria-specific reagents such as MitoSOX to detect the effects of quercetin on mitoROS according to your suggestion, so as to confirm our relevant speculation. Thank you again for your constructive comments on our study. Thank you.

1. Fock, E. M., Parnova, R. G. (2021). Protective effect of mitochondria-targeted antioxidants against inflammatory response to lipopolysaccharide challenge: a review. Pharmaceutics. 13 (2), 144. doi: 10.3390/pharmaceutics13020144

Round 2

Reviewer 2 Report

The authors have introduced relevant changes according to my suggestions. I accept the revised version of the manuscript. 

Reviewer 3 Report

The author modify some problems can be suitable for the quality of IJMS